# Phytochemical Composition, Antioxidant, Anti-*Helicobacter pylori*, and Enzyme Inhibitory Evaluations of *Cleistocalyx operculatus* Flower Bud and Leaf Fractions

**DOI:** 10.3390/biotech13040042

**Published:** 2024-10-11

**Authors:** Doan Thien Thanh, Mai Thanh Tan, Nguyen Thi My Thu, Pham Nhat Phuong Trinh, Pham Thi Hoai Thuong, Pham Thi Giang Tuyet, Luong Thi My Ngan, Tran Trung Hieu

**Affiliations:** 1Faculty of Biology and Biotechnology, VNUHCM-University of Science, Ho Chi Minh City 700000, Vietnam; doanthienthanh@tdtu.edu.vn (D.T.T.); thanhtan.mtt1996@gmail.com (M.T.T.); nguyenmythu2112@gmail.com (N.T.M.T.); 2Faculty of Applied Sciences, Ton Duc Thang University, Ho Chi Minh City 700000, Vietnam; pnhatptrinh@gmail.com (P.N.P.T.); ptht.011001@gmail.com (P.T.H.T.); giangtuyet2605@gmail.com (P.T.G.T.)

**Keywords:** *Cleistocalyx operculatus*, antioxidant effect, anti-*H. pylori* activity, antibiofilm formation, membrane permeability, morphological transformation

## Abstract

Six solvent fractions isolated from flower bud and leaf ethanolic extracts of *Cleistocalyx operculatus* were analyzed for their phytochemical contents, including phenolics, flavonoids, saponins, tannins, and alkaloids. Antioxidant activities were measured using the ABTS, DPPH, and FRAP assays. The results showed that the flower bud aqueous fraction (BAF) and the leaf aqueous fraction (LAF) rich in phenolic content (768.18 and 490.74 mg GAE/g dry extract, respectively) exhibited significantly higher antioxidant activities than the other fractions. The flower bud hexane fraction (BHF) had remarkably high flavonoid and saponin contents (134.77 mg QE/g and 153.33 mg OA/g dry extract, respectively), followed by that of the leaf hexane fraction (LHF) (76.54 mg QE/g and 88.25 mg OA/g dry extract, respectively). The BHF and LHF were found to have extremely high antibacterial activity against two *H. pylori* strains, ATCC 51932 and 43504 (MICs of 125 µg/mL). Interestingly, DMC (2′,4′-Dihydroxy-6′-methoxy-3′,5′-dimethylchalcone) isolated from the BHF displayed greater antibacterial activity against the bacterial strains (MICs of 25–50 µg/mL) than those of the fractions. In addition, DMC presented potent inhibitory effects on *H. pylori* urease (IC_50_ of 3.2 µg/mL) and α-amylase (IC_50_ of 83.80 µg/mL), but no inhibition against α-glucosidase. It was also demonstrated that DMC showed pronounced inhibitory effects on the urease activity and biofilm formation of *H. pylori*, and could increase the membrane permeability of the bacterial cells. Scanning electron micrographs depicted that the BHF and DMC had strong effects on the cell shape and significantly induced the distortion and damage of the cell membrane. The fractions and DMC showed no significant toxicity to four tested human cell lines. Efforts to reduce antibiotic use indicate the need for further studies of the flower buds and DMC as potential products to prevent or treat gastric *H. pylori* infections.

## 1. Introduction

The flower buds and leaves of *Cleistocalyx operculatus* (Roxb.) Merr. and L.M. Perry (or *Syzygium nervosum* DC.) are rich sources of bioactive compounds, predominately containing flavonoids, chalcones, and triterpenoids [1,2]. In Asian cultures, these herbal materials have been traditionally used as tea or remedies in the treatments of various disorders, such as influenza, bacillary dysentery, gastric inflammation, abdominal pain, and skin infection, as well as for their antiseptic properties [3,4]. Several solvent extracts of the flower buds have been known to exert various pharmacological activities in vitro and in vivo, including anti-hyperglycemic and cardio-tonic effects [5,6,7,8]. DMC (2′,4′-Dihydroxy-6′-methoxy-3′,5′-dimethylchalcone), a major constituent of the flower buds, was found to significantly inhibit the growth of human liver cancer cells and human umbilical vein endothelial cells [2,9]. Few works refer to the antibacterial activities of DMC, particularly towards *H. pylori*. DMC was reported to inhibit the growth of *Bacillus subtilis*, *Escherichia coli*, and *Cladosporium cucumerinum* [10]. A methanolic extract of the leaves was shown to have inhibitory activity against Gram-positive bacteria (e.g., *Staphylococcus aureus*, *Bacillus subtilis*, and *Streptococcus mutans*) and the yeast *Candida maltosa*, but no effect on Gram-negative bacteria (e.g., *Escherichia coli* and *Pseudomonas aeruginosa*) [6]. However, previous research indicated that the crude ethanolic extract of the leaves showed high antibacterial activity against *Helicobacter pylori* ATCC 51932 and three clinical isolates of *H. pylori* [11]. Recent research has reported that the crude hexane extract of *C. operculatus* flower buds exhibited the most growth-inhibitory activity against *Salmonella typhimurium* and *H. pylori*, while the crude ethanol and methanol extracts of the flower buds exhibited the strongest antioxidant activities [4]. Furthermore, the hexane extract was found to have the strongest inhibitory effect on *H. pylori* urease activity [11].

*H. pylori* is a Gram-negative bacterium with a curved or spiral shape that infects and colonizes the human gastric mucosa [12]. The bacterial infections are prevalent in over half of the world’s population, particularly in developing regions, where the infections can affect as much as 90% of the population and tend to endure for a person’s lifetime [13]. *H. pylori* infections cause asymptomatic chronic active gastritis in most infected individuals and lead to stomach ulcer disease and mucosa-associated lymphoid tissue lymphoma [14,15]. Several important factors, including vacuolating cytotoxin A, cytotoxin-associated gene A, chemotactic motility, adhesins, biofilm formation, and urease production of *H. pylori*, are known to contribute to the virulence of this organism [16,17,18]. Its spiral shape is the predominant form involved in the ability to thrive and colonize the gastric epithelial cells. Under unfavorable conditions, the spiral forms can convert into coccoid forms as a survival mechanism, but the coccoid forms are much less infective and virulent, and less likely to colonize and cause inflammation [18]. In addition, the biofilm formation of *H. pylori* provides the bacterium with protection and resistance to antimicrobial agents [19]. Nowadays, global antibiotic resistance in *H. pylori* is on the rise in many parts of the world, leading to treatment failure and reinfection [12].

This study aims to investigate the phytochemical contents in six solvent fractions of the *C. operculatus* flower buds and leaves, and evaluate their antioxidant, enzyme inhibitory, and anti-*H. pylori* activities. The effects of the fractions and DMC on biofilm formation, membrane permeability, and the cell morphology of *H. pylori* were also evaluated.

## 2. Materials and Methods

Reagents, chemicals, and enzymes utilized in the bioactivity studies were purchased from Sigma-Aldrich (Schnelldorf, Germany) and Merck (Damstadt, Germany). Two *Helicobacter pylori* strains (ATCC 51932 and ATCC 43504), MCF-7 cells (HTB-22, human breast cancer cell line), Jurkat cells (TIB-152, blood cancer cell line), and HeLa cells (CCL-2, cervical carcinoma cell line) were provided by ATCC (The American Type Culture Collection, Manassas, Rockville, MD, USA). Fibroblast cells derived from human foreskins were provided by the Laboratory of Molecular Biology, Department of Genetics, VNUHCM-University of Science [20]. Media and serum were obtained from HIMEDIA (Maharashtra, India), Becton, Dickinson and Company (Franklin Lakes, NJ, USA), Sparks (Aukland, New Zealand), Gibco-Thermo Fisher Sciemtific (Auckland, New Zealand), and Sigma-Aldrich (St. Louis, MI, USA). All of the solvents and other chemicals were of reagent grade and commercially available.

### 2.1. General Experimental Procedures

An Agilent 1260 infinity series HPLC system (San Jose, CA, USA) with a ZORBAX Eclipse Plus C18 column (4.6 × 150 mm, 3.5 μm) was used, which was maintained at 25 °C with a UV detection at 320 nm. The mobile phase compositions were acetonitrile and 0.1% formic acid (70:30, *v*/*v*) with a flow rate of 1 mL/min. The injection volume was 10 μL with a running time of 75 min. The ^1^H and ^13^C NMR (Nuclear Magnetic Resonance) spectra were obtained in CDCl_3_ at 25 °C using a 500 MHz Bruker AVANCE III HD spectrometer (Ettlingen, Germany). The Fourier transform infrared *spectroscopy* (FT-IR) spectrum was recorded by a Bruker Tensor 27 FT-IR Spectrometer (Bruker Optik GmbH, Ettlingen, Germany). Silica Gel 60 (0.06–0.2 mm) (Scharlab, Sentmenat, Spain) was used in a silica gel column (∅2 × 30 cm).

### 2.2. Extraction and Isolation of Plant Material

Dried flower buds and leaves of *C. operculatus* were collected from the traditional medicine and herbal material market on Hai Thuong Lan Ong Street, Dist. 5th, Ho Chi Minh City, Vietnam, and identified at the Department of Plant Biotechnology and Biotransformation, Faculty of Biology and Biotechnology, VNUHCM-University of Science, Ho Chi Minh City, Vietnam. Herbarium specimens of the flower buds and leaves were deposited in the Department under the code COFB1001 [4] and COL1002, respectively. Ethanolic crude extracts of the flower buds and leaves were prepared following the method described by Thanh et al. (2024) [4]. The extracts of the flower buds (90 g) and leaves (81 g) were partitioned by liquid–liquid fractionation to yield 1.8 g of flower bud hexane fraction (BHF), 78 g of flower bud ethyl acetate fraction (BEF), and 10.2 g of flower bud aqueous fraction (BAF), and 26.7 g of leaf hexane fraction (LHF), 24.3 g of leaf ethyl acetate fraction (LEF), and 30 g of leaf aqueous fraction (LAF), respectively, which are presented in the Appendix A.

Guided by the anti-*H. pylori* tests, the BHF (1.8 g), one of the most active fractions, was purified by silica gel column chromatography eluted with an *n*-hexane (H) and ethyl acetate (EA) (100:0–0:100) gradient to obtain six column fractions, fraction 1 (34.7 mg), fractions 2–3 (17.6 mg), and fractions 4–6 (56.4 mg), described previously by [9] (Appendix A). Repetitive column chromatography of fractions 2–3 (17.6 mg) eluted with H/AE = 90:10 (Appendix A) resulted in an active principle 1 (14.2 mg) as yellow needle-shaped crystals.

The purity of principle 1 was found to be greater than 95% by the HPLC analysis (Appendix A), and its spectroscopic data matched the data reported in previous research by Choommongkol et al. (2022) [21]. The principle 1, DMC (2′,4′-Dihydroxy-6′-methoxy-3′,5′-dimethylchalcone), was identified based on the following evidence: FT-IR ν (KBr): the stretching vibrations of the O-H (3421 cm^−1^), ν_Ar-H_ and ν_=C–H_ (3001–3028 cm^−1^), the stretching vibrations of the C–H (2855–2925 cm^−1^), the C=O (1626 cm^−1^), the C_Ar_=C_Ar_ (1451–1540 cm^−1^), the deformation vibrations of the C-H (1359–1419 cm^−1^), the stretching vibrations of the C-O (1111–1220 cm^−1^), and the out-of-plane deformation vibrations of the C_Ar_-H and =C–H (612–987 cm^−1^) (Appendix A). The ^1^H NMR, ^13^C NMR, and HMBC data of DMC are presented in Appendix A. The spectral data were identical to the published data of Choommongkol et al. (2022) [21]. Additionally, the ^1^H–^13^C HMBC-correlations of DMC are presented in Appendix A.

### 2.3. Total Phytochemical Contents

The total phenolic content (TPC) was quantified using the Folin–Ciocalteu assay, as indicated by Temesgen et al. (2022) [22]. An aliquot of each fraction (0.9 mL) and 4.5 mL of Folin–Ciocalteu reagent (10%) was vigorously shaken and incubated for 5 min in the dark. Then, 1.8 mL of Na_2_CO_3_ (7%) was added into the mixture and continued to incubate for 30 min. Absolute ethanol was used as a control (blank), and gallic acid (0–100 µg/mL) was also prepared to build a standard curve (y = 0.0095x + 0.0029; R^2^ = 0.9978). Absorbance at 765 nm was subsequently measured using a spectrophotometer (UV-5100, Metash, Shanghai, China). The TPC was conveyed as the mg gallic acid equivalent (GAE)/g of dry fraction.

The total flavonoid content (TFC) was determined following the aluminum chloride colorimetric method [23]. An aliquot of each fraction (300 µL) was mixed with 150 µL of NaNO_2_ solution (5%) and incubated for 6 min at room temperature. The mixture was then added with 300 µL of AlCl_3_ (5%) and 1 mL of NaOH 1 M and allowed to stand for 6 min. Absolute ethanol was prepared as a control (blank) to replace the extract. Quercetin (0–100 µg/mL) was used to estimate the standard curve (y = 0.0075x − 0.0017; R^2^ = 0.9998). The absorbance of the mixture was measured at 510 nm. The TFC was stated as the mg of quercetin equivalent (QE)/g of dry fraction.

The total alkaloid content (TAC) was quantified following the modified method of Ncube et al. (2015) [24]. Briefly, 1 mL of each fraction diluted in HCl (2N) was mixed well with 5 mL of bromocresol green (BCG) solution (0.01%) and 5 mL of phosphate buffer solution at a pH of 4.7 in a separating funnel. The mixture was vigorously shaken and extracted twice with 5 mL of chloroform. Then, the collected alkaloid extracts were diluted with 10 mL of chloroform. The absorbance of the mixture was read at 470 nm. Atropine (0–100 µg/mL) and absolute ethanol were used as the standard solution (y = 0.0128x − 0.0493; R^2^ = 0.9927) and control solution, respectively. The total content of alkaloids was expressed as the mg atropine equivalent (AE)/g of dry fraction.

The total tannin content (TTC) was quantified using a previously reported method [23]. An aliquot of each fraction (0.5 mL) was added into 3.0 mL of vanillin solution (4% in methanol, *w*/*v*). Then, the mixture was stirred with 1.5 mL of HCl and incubated for 15 min in the dark. The absorbance of the mixture was measured at 500 nm, and absolute ethanol was the control (blank) to replace the extract. The TTC was calculated using the catechin standard curve (y = 0.0012x + 0.0261; R^2^ = 0.9984) and shown as the mg of catechin equivalent (CE)/g of dry fraction.

The total saponin content (TSC) was determined using the vanillin method [25]. Briefly, each fraction (0.5 mL) was mixed with 0.1 mL of vanillin solution (5% in acetic acid, *w*/*v*) and 0.4 mL of 70% perchloric acid. The mixture was heated at 60 °C for 15 min in a water bath and then cooled to room temperature. After that, 5 mL of absolute acetic acid was mixed well into the mixture. Oleanolic acid (0–300 µg/mL) was used to measure the standard curve (y = 0.0041x − 0.0256; R^2^ = 0.9934). The absorbance values were measured at 548 nm. The TSC was expressed as the mg oleanolic acid equivalent (OAE)/g of dry fraction.

### 2.4. Antioxidant Assay

A DPPH (2,2-diphenyl-1-(2,4,6-trinitrophenyl)hydrazyl) radical scavenging assay was followed according to a previous study, as described by Elouafy et al. (2023) [26], to assess the antioxidant activity. Briefly, different concentrations of each fraction (0–150 μg/mL) were prepared in methanol. Each dilution (1 mL) was well mixed with 1 mL of DPPH methanolic solution (25 µg/mL) and incubated at 37 °C in the dark for 30 min. The absorbance of the mixture was subsequently measured at 517 nm. Ascorbic acid and absolute methanol were used as the positive control and blank, respectively. The DPPH scavenging activity was measured using Equation (1), as follows:(1)$$DPPH\;scavenging\;activity\;(\%)=\frac{{OD}_{1}-{OD}_{2}}{{OD}_{1}}\times 100\;(\%)$$
where OD_1_ and OD_2_ are the absorbance values of the blank and each fraction or ascorbic acid, respectively. The concentration required to inhibit 50% of the free DPPH radial (IC_50_) was calculated by plotting the DPPH scavenging activity versus the sample concentration.

The measurement of the antioxidant activity was also performed using the ABTS•+ (2,2′-azinobis(3-ethylbenzothiazoline-6-sulfonic acid) diammonium salt) assay presented by Olszowy-Tomczyk and Typek (2024) [27], with some modifications. Briefly, the initial solutions consisted of 7 mM ABTS•+ solution (5 mL) and 140 mM K_2_S_2_O_8_ solution (88 μL). The working solution was reacted in the dark at room temperature for 16 h and then diluted in ethanol until the absorbance value of 0.7 ± 0.002 at 734 nm using a spectrophotometer. Fresh ABTS•+ solution was created for each assay. For the analysis, each fraction (0.01 mL) was mixed with 0.99 mL of the ABTS•+ solution. The absorbance of the mixture was then measured at 734 nm after 6 min of storage in the dark. After 6 min of storage in the dark, the absorbance was measured at 734 nm. Absolute methanol and Trolox (6-hydroxy-2,5,7,8-tetramethylchroman-2-carboxylic acid) were used as the negative control and positive control, respectively. The ABTS•+ scavenging capacity was evaluated as the percentage of inhibition of the ABTS radical scavenging activity using Equation (2), as follows:(2)$$ABTS\;radical\;scavenging\;activity\;(\%)=\frac{{OD}_{1}-{OD}_{2}}{{OD}_{1}}\times 100\%$$
where OD_1_ and OD_2_ are the absorbance of the ABTS•+ radical cation in methanol and in each fraction or Trolox, respectively. The concentration required to inhibit the 50% ABTS•+ solution (IC_50_) was subsequently estimated by plotting the ABTS radical scavenging activity versus the sample concentration.

The FRAP assay was established using a modified method [23]. Briefly, the FRAP reagent was prepared by mixing 300 mM sodium acetate buffer at a pH of 3.6 (25 mL), 10 mM TPTZ (2, 4, 6- tripyridyl-s-triazine) solution in 40 mM HCl (2.5 mL), and 20 mM FeCl_3_.6H_2_O solution (2.5 mL), and then warmed at 37 °C before use. An amount of 150 μL of each fraction was mixed with 2850 μL of the FRAP reagent and then incubated at 37 °C for 30 min in the dark in order to initiate the reaction. The standard curve of the assay was also prepared by using a serial concentration of Trolox (0–500 µg/mL). The absorbance of the colored product was read at 593 nm. The antioxidant capacity was stated as the mg Trolox equivalent (TE)/g of the dry fraction.

### 2.5. Anti-Helicobacter pylori Assay

Two *H. pylori* strains were stored in BHI (brain heart infusion) broth medium supplemented with 10% NBS (newborn bovine serum) containing vancomycin (10 mg/L), polymyxin B (5 mg/L), trimethoprim (5 mg/L), and amphotericin B (2 mg/L), and 25% glycerol in a liquid nitrogen container until use. The minimal inhibitory concentrations (MICs) of the fractions and DMC were established using a broth dilution assay in sterile 96-well plates for the bacterial strains [28]. Amount of 25 μL of the tested samples at different concentrations from 1 to 1000 μg/mLdiluted in DMSO was added to 75 μL of Brucella broth supplemented with 10% NBS.The final concentration of the DMSO in the assay was less than 2.5%. Generally, 30 μL of bacterial suspension (5 × 10^6^ CFU/mL) of each strain was added to 10 μL of tested samples. Then, the plates were incubated at 37 °C and shaken at 50 rpm for 48 h in a microaerophilic condition using a Oxoid CampyGen sachet (Thermo Fisher Scientific, Loughborough, UK) placed in a sealed jar. The MIC value was defined as the lowest concentration that visibly inhibited bacterial growth using resazurin as an indicator.

### 2.6. Enzyme Inhibitory Assay

Crude urease of *H. pylori* ATCC 43504 was prepared according to the method of Ngan et al. (2012) [29]. Briefly, 10 μL of each fraction or DMC at various concentrations of 0–500 μg/mL was added in 30 μL of 20 mM EDTA–sodium phosphate buffer (pH of 7.3). Then, 10 μL of urease solution (75 μL of urease/1 mL of the buffer) was added and incubated at room temperature for 1 h before adding 50 μL of urea solution (0.24 mg of urea/1 mL of the buffer), and allowed to incubate at room temperature for 30 min. After that, 40 μL of solution A (40% sodium salicylate and 0.3% sodium nitroprusside) and 60 μL of solution B (0.5% sodium hydroxide and 0.042% sodium hypochlorite) were added into the mixture. The ammonia generated by the urease activity was quantified by measuring the absorbance on a Microlisa Plus microplate reader (Micro Lab Instruments, Ahmedabad, India) at 625 nm with ammonium chloride as a standard and buffer solution as a control.

The α-glucosidase activity was determined by measuring the release of p-nitrophenol from pNPG (p-nitrophenyl-α-D-glucopyranoside) according to the method of Shai et al. (2011) [30], with slightly modification. A serial concentration (0–500 μg/mL) of each fraction and DMC were dissolved in DMSO (dimethyl sulfoxide). A mixture containing 40 µL of tested samples, 20 µL of α-glucosidase (2.0 U/mL), and 100 µL of potassium phosphate buffer (100 mM, pH = 6.8) was prepared and preincubated at 37 °C for 15 min. The reaction was initiated by adding 40 µL of pNPG 5 mM and incubated at 37 °C for 20 min. To stop the reaction, 100 µL of Na_2_CO_3_ 0.1 M was added to the solution. The amount of p-nitrophenol released by α-glucosidase was defined as measuring the absorbance at 405 nm. Acarbose was used as the standard inhibitor. The control solution contained buffer solution (40 μL) instead of the fractions, DMC, or standard inhibitor.

The α-amylase inhibitory activity of the fractions and DMC were caried out according to the method of Ogunyemi et al. (2022) [31], with slightly modifications. A mixture of 100 μL of various concentrations (0–500 μg/mL in DMSO) of each fraction or DMC and 100 μL of sodium phosphate buffer (0.02 M, pH of 6.9) containing 200 μL of α-amylase solution (2.5 U/mL) were incubated at room temperature for 10 min. After preincubation, the mixture was added with 200 μL of 1% soluble starch solution in sodium phosphate buffer. The mixture was reacted at 30 °C for 10 min before adding of 300 μL of HCl 1 N to stop the reaction. The mixture was added to 90 μL of 10% iodine solution to observe the color change and diluted with 10 mL of distilled water. The sodium phosphate buffer and acarbose served as the control and standard inhibitor, respectively. The absorbance was read at 540 nm.

These enzyme inhibitory activities were determined using Equation (3), as follows:(3)$$Inhibition\;activity\left(\%\right)=\frac{\left(Ac-As\right)}{Ac}\times 100\%$$
where Ac is the absorbance of the control and As is the absorbance of the fractions, DMC, or standard inhibitors.

### 2.7. Biofilm Formation Inhibitory Assay

The antibiofilm activity of the fractions and DMC was carried out in 96-well plates following the method of Hieu et al. (2022) [32]. In brief, 30 μL of *H. pylori* ATCC 43504 suspension (10^8^ CFU/mL) and 10 μL of the tested samples at sub-MICs (MIC/2, MIC/4, and MIC/8) were added into 60 μL of Brucella broth in each well of sterile 96-well plates. Blank wells containing DMSO and background wells containing samples were also similarly arranged as control wells without the bacterial suspension. After 48 h of incubation at 37 °C in a microaerobic environment with shaking at 150 rpm, the medium was removed by using PBS (0.01 M phosphate-buffered saline, pH of 7.2) to wash the wells. The plates were then air-dried before being fixed with absolute methanol for 15 min and dried. Each well was dyed with a 0.1% crystal violet solution for 10 min and washed with distilled water. The dried plates were dissolved with crystal violet using 95% ethanol for 15 min. The Microlisa Plus microplate reader was used to measure the absorbance at 595 nm.

### 2.8. Scanning Electron Microscopy

To determine the efficacy of the BHF and DMC against *H. pylori* through the morphological changes, a scanning electron microscopy (SEM) analysis was performed [33]. Overnight broth cultures of *H. pylori* ATCC43504 were prepared in Brucella broth. The cell suspension of *H. pylori* was cultured after 48 h with or without tested samples at MIC concentrations. Then, the bacteria were harvested by centrifugation at 14,000 rpm for 5 min. Briefly, the specimens were fixed in modified Karnovsky’s fixative (2% glutaraldehyde and 2% paraformaldehyde in 0.05 M sodium cacodylate buffer, pH of 7.2) for 4 h, and then post-fixed in 1% osmium tetroxide in 0.05 M sodium cacodylate buffer for 4 h at 4 °C. A series of increasing concentrations of ethanol (20, 40, 60, 80, 95%, and absolute) and hexamethyldisilazane were used to dehydrate the specimens, each twice for 15 min. SEM was performed by the Institute of Chemical Technology, Ho Chi Minh City, Vietnam. The specimens were then mounted on SEM stubs by double-sided carbon conductive tape and coated with gold (JEC-3000FC ion sputter, JEOL Co., Tokyo, Japan). The SEM images were obtained using a scanning electron microscope, the JSM-IT200 (JEOL Co., Japan), operating at an acceleration voltage of 5 kV.

### 2.9. Membrane Permeability Assay

The membrane permeability of *H. pylori* was determined by a crystal violet assay [34]. Briefly, 30 μL suspensions of *H. pylori* ATCC 43504 (10^8^ CFU/mL) were prepared in 60 μL of BB medium and 10 μL of each fraction and DMC at different concentrations (MIC, MIC/2, MIC/4, and MIC/8). The mixture was incubated for 2 h and the cells were harvested at 4500 rpm for 5 min at 4 °C. The cells were washed twice in PBS (0.01 M, pH of 7.2). After that, the cells were resuspended in 1 mL of PBS containing 100 μL of 0.1% crystal violet and incubated for 5 min at 37 °C. The suspension was then centrifuged at 14,000 rpm for 15 min and the supernatant was obtained with the Microlisa Plus microplate reader at 590 nm. The percentage of the crystal violet uptake of the *H. pylori* cells in all of the tested samples was calculated using Equation (4), as follows:(4)$$Crystal\;violet\;uptake\left(\%\right)=\frac{\left(1-OD\;value\;of\;the\;sample\right)}{OD\;value\;of\;crystal\;violet\;solution}\times 100\%$$

### 2.10. Cell Line Cultures and Cytotoxicity Assay

MCF-7, HeLa, Jurkat, and fibroblast cell lines were grown in EMEM (Eagle’s minimal essential medium) for the MCF-7 and HeLa cells, RPMI (*Roswell Park Memorial Institute*) medium for the Jurkat cells, and DMEM/F12 (Dulbecco’s modified eagle medium: nutrient mixture F12) medium for the fibroblast cells. These media were supplemented with 10% FBS (fetal bovine serum), 2 mM L-glutamine, 20 mM HEPES, 0.025 μg/mL of amphotericin B, 100 IU/mL of penicillin G, and 100 μg/mL of streptomycin at 37 °C and 5% CO_2_. The 2nd to 5th passages of the cultivations of the fibroblast cells, and the 4th to 20th passages of the cultivations of the HeLa, MCF7, and Jurkat cells, were used.

The SRB (Sulforhodamine B) assay was performed according to Nguyen and Huynh (2016) [20]. In brief, the cells were seeded in 96-well plates at a density of 10,000 cells/well for the MCF-7, HeLa, and fibroblast cells, and 50,000 cells/well for the Jurkat cells. These cells were then cultured for 24 h before exposure to varying concentrations of each fraction or DMC for a duration of 48 h. Treated cells were fixed with a cold 50% (*w*/*v*) trichloroacetic acid solution for 1–3 h, then washed and stained with 0.2% (*w*/*v*) SRB for 20 min. After washing with 1% acetic acid five times, protein-bound dye was solubilized in a 10 mM Tris base solution. Optical density values were measured using the Microlisa Plus microplate reader at wavelengths of 492 nm and 620 nm. Camptothecin and 0.25% DMSO were used as the positive and negative controls, respectively. The percentage of growth inhibition was determined using Equation (5), as follows:(5)$$Inhibition\left(\%\right)=1-\frac{ODt}{ODc}\times 100\%$$
where ODt is the optical density value of the tested sample and ODc is the value of the control sample.

### 2.11. Statistical Analysis

The MIC values of each test fraction and compound were established with at least three independent experiments performed in triplicate (n ≥ 9). Tested materials with MIC values of ≤130, >130–<630, 630–1250, >1250–<2500, and ≥2500 µg/mL were classified as extremely high, high, moderate, low, and no inhibitory activity against bacteria growth, respectively [35]. All other experiments were performed in triplicate and the data are shown as mean ± standard derivations (SD) (n ≥ 3). Analysis of variance (ANOVA) by Tukey’s multiple comparison, tested as *p* < 0.05, and the half-maximal inhibitory concentration value (IC_50_) and half-cytotoxicity concentration (CC_50_) value were conducted using the GraphPad Prism 8 software program (San Diego, CA, USA).

## 3. Results

### 3.1. Total Phytochemical Contents

The total phytochemical contents of different fractions from the flower buds and leaves of *C. operculatus* are shown in Table 1. The BAF and LAF revealed the highest values of the TPC, with 768.18 and 490.74 mg GAE/g dry extract, respectively. The lowest TPC was found in the LHF (201.63 mg GAE/g dry extract). In contrast, the BHF showed the highest TFC and TSC (134.77 mg QE/g dry extract and 153.33 mg OA/g dry extract, respectively), followed by the BEF (85.88 mg QE/g dry extract and 158.10 mg OA/g dry extract, respectively) and the LHF and LEF (71.72–76.54 mg QE/g dry extract and 75.24–88.25 mg OA/g dry extract, respectively). The LAF contained the lowest levels of both the TFC and TSC (25.56 mg QE/g dry extract and 25.56 mg OA/g dry extract,) while the BAF had the lowest level of the TFC (11.04 mg QE/g dry extract). In the experiment, the flower bud and leaf fractions of *C. operculatus* presented relatively low values of the TTC (4.36–42,97 mg CE/g dry extract) and significantly low values of the TAC (1.50–5.41 mg AE/g dry extract).

### 3.2. Antioxidant Activity

The antioxidant activities of the different fractions from the flower buds and leaves of *C. operculatus* are exhibited in Table 2. Depending on the solvent used for extracting the fractions, the antioxidant activity of the fractions significantly varied. In the FRAP assay, the LAF exhibited the highest antioxidant activity (301.82 mg TE/g dry extract), followed by the BAF (201.80 mg TE/g extract). The lowest FRAP activity was obtained with the LHF (55.57 mg TE/g extract) and BHF (69.01 mg TE/g extract). The result of the DPPH and ABTS radical scavenging abilities indicated that the LAF exhibited the strongest antioxidant activity (IC_50_ values of 11.24 and 0.55 μg/mL, respectively) followed by the LEF (16.05 and 0.98 μg/mL) and BAF (24.69 and 1.08 μg/mL). However, the result from the ABTS method showed that the LEF, LHF, and BAF provided similar ABTS radical scavenging activity (0.98–1.08 μg/mL). Weak DPPH radical scavenging activity was found in the BHF, LHF, and BEF (IC_50_ range of 33.99–58.46 μg/mL), while weak ABTS radical scavenging activity was found in the BEF and BHF (1.48–1.70 μg/mL). All of the tested fractions produced lower antioxidant effects than that of ascorbic acid (IC_50_ of 3.34 μg/mL) but higher than that of the Trolox (IC_50_ of 2.63 μg/mL).

### 3.3. Anti-H. pylori Activity

The bacterial growth inhibitory activities of the *C. operculatus* fractions and DMC against two *H. pylori* strains (ATCC 51932 and ATCC 43504) are shown in Table 3. Both the BHF and LHF showed extremely high growth inhibitory activity against the both strains, with an MIC value of 125 µg/mL. Other fractions displayed a high growth inhibitory effect, with MIC values ranging from 250 to 500 µg/mL on both bacterial strains, except the LAF, with a moderate effect (an MIC value of 1000 µg/mL) on the strain 43504. Remarkably, DMC revealed stronger antibacterial activity (MIC values of 25 and 50 µg/mL) than those of the BHF and LHF. The amoxicillin MIC value against the strains was 0.01 µg/mL. This indicated that *H. pylori* ATCC 51932 and ATCC 43504 were susceptible amoxicillin strains, as the MIC resistance breakpoints for amoxicillin against *H. pylori* was reported to be >0.5 µg/mL [35].

### 3.4. Enzyme Inhibitory Activity

The inhibitory activities of the *C. operculatus* fractions and DMC against *H. pylori* urease, α-glucosidase, and α-amylase are presented in Table 4. All of the tested fractions and DMC exhibited strong inhibition against *H. pylori* urease (IC_50_ values ranging from 2.3 to 3.6 μg/mL), except for the BEF (IC_50_ of 4.9 μg/mL) and LAF (IC_50_ of 6.8 μg/mL). These fractions and DMC showed better inhibitory activity against urease than that of thiourea (IC_50_ of 44.3 μg/mL) as a positive control. These fractions also had a strong inhibitory effect on α-glucosidase (IC_50_ values ranging from 0.6 to 2.6 μg/mL) and significantly stronger than the positive control, acarbose (IC_50_ of 25.6 μg/mL), whereas DMC had no inhibition against α-glucosidase (IC_50_ of 94.6 μg/mL). Nevertheless, these fractions exhibited weak (IC_50_ values ranging from 191.3 to 497.2 μg/mL) and no (IC_50_ of 1281.7 μg/mL) inhibitory activities against α-amylase compared with acarbose and DMC, which had similar inhibitory effects (IC_50_ values of 75.0 and 83.8 μg/mL, respectively).

### 3.5. Effect on Biofilm Formation of H. pylori

The results of the biofilm inhibition revealed that all of the tested fractions and DMC at sub-MICs inhibited the biofilm formation of *H. pylori* after 48 h of treatments (Figure 1). At MIC/2, all of the fractions (62.5–500 µg/mL) and DMC (25 µg/mL) demonstrated potential antibiofilm activity and reduced biofilm formation by 82.2 ± 0.74–93.6 ± 1.53%, with a nonsignificant difference (*p* > 0.05). At MIC/4, the LEF (125 µg/mL) and DMC (12.5 µg/mL) induced pronounced antibiofilm effects, with inhibitory percentages of 78.4 ± 0.98 and 77.8 ± 1.37%, respectively, followed by the slightly lower activity of the BHF (31.25 µg/mL) and BEF (125 µg/mL), reducing the biofilm formation by 73.5 ± 0.98% and 70.2 ± 2.55%, respectively. However, the LHF (31.25 µg/mL), BAF (125 µg/mL), and LAF (250 µg/mL) reduced biofilm formation by 50.9 ± 6.86%, 54.3 ± 2.43%, and 57.8 ± 0.88%, respectively. Notably, at MIC/8, the BEF and LEF (62.5 µg/mL) remained able to inhibit biofilm formation by 58.2 ± 0.59% and 57.1 ± 0.98%, respectively, followed by DMC (6.25 µg/mL), which inhibited biofilm formation by 46.6% ± 0.49%. Other fractions had weak inhibitory effects (21.1 ± 0.49–36.4 ± 2.84%).

### 3.6. Effect on the Morphology of H. pylori

The scanning electron micrographs for the untreated *H. pylori* showed that the cells appeared mainly in a spiral-shaped form that successfully incorporated within the biofilms (Figure 2A,B). The cell surface was smooth and regular with an intact cell membrane. However, after 48 h of treatment with the BHF at an MIC = 125 µg/mL (Figure 2C,D) and DMC at an MIC = 50 µg/mL (Figure 2E,F), 86.2 ± 3.56 and 93.7 ± 3.68% of the spiral cells were converted into coccoid-shaped cells, respectively. These figures indicated that the BHF and DMC could inhibit bacterial biofilm formation and induce clusters of coccoid cells. The clusters were aggregated and stuck to each other with rough or deformed membrane surfaces and cell debris. This suggests that the majority of coccoid forms induced by the BHF and DMC revealed the morphological manifestations of bacterial cell death or destruction.

### 3.7. Effect on Membrane Permeability

The alteration in the membrane permeability of *H. pylori* caused by *C. operculatus* fractions and DMC is presented in Figure 3. The crystal violet uptake by untreated cells was only by 13.8 ± 1.04–26.0 ± 0.98%, but the crystal violet uptake by treated cells increased from 63.0 ± 0.87 to 71.3 ± 1.46%, which was caused by all of the tested fractions, and 79.8 ± 1.06%, which was caused by DMC at MIC/2 after 2h of treatment. At MIC/4, DMC (12.5 µg/mL) induced an extreme increase in the crystal violet uptake by 64.4 ± 2.76%, followed by those of the LHF and BHF (31.25 µg/mL), LEF and BAF (125 µg/mL), and LAF (250 µg/mL), increasing by 49.5 ± 2.43% and 50.3 ± 1.54%, 46.3 ± 4.17% and 54.8 ± 2.49%, and 50.1 ± 1.36%, respectively, and with the BEF (125 µg/mL), increasing by only 33.82 ± 1.73%. At MIC/8, the membrane permeability of the bacterial cells did not alter much (27.4 ± 1.90–40.4 ± 2.06%) compared with those of the untreated groups.

### 3.8. Cytotoxicity Effects

In order to investigate the selectivity of the anti-*H. pylori* activity of *C. operculatus* fractions and DMC, the cytotoxic activity of these tested samples on four human cell lines was examined, as shown in Table 5. All of the fractions and DMC had no significant cytotoxicity against fibroblast cells, with CC_50_ values of >100 µg/mL. Among these factions, the BAF exhibited no cytotoxic activity against the four tested cell lines (CC_50_ values > 100 µg/mL), whereas the BHF displayed the most cytotoxic effect on three cancer cell lines, Jurkat, MCF-7, and HeLa cells, with CC_50_ values of 18.51, 30.79, and 31.70 µg/mL, respectively. Other fractions and DMC had no cytotoxicity to the three cancer cell lines (CC_50_ values ranging from 85.43 to >100 µg/mL) or weak cytotoxicity (CC_50_ values ranging from 51.06 to 77.07 µg/mL). However, the four tested cell lines were less sensitive to these fractions and DMC when compared with the positive control camptothecin, with a CC_50_ of 0.005–1.57 µg/mL (*p* < 0.001).

## 4. Discussion

Phytochemicals (e.g., polyphenols, flavonoids, alkaloids, tannins, and saponins) are known to play an important role in overall health and disease prevention. In the *C. operculatus* flower bud and leaf fractions, the contents of phytochemicals significantly varied depending on the solvents used. Similarly with other previous studies, a high polyphenol content was mostly found in the aqueous extracts of both *C. operculatus* flower buds and leaves [22,23,37,38,39]. Minh et al. (2023) [40] recently showed that the flavonoid content was only present at a low level in the aqueous extracts of *C. operculatus* fresh leaves (2.73 mg QE/g) when compared with that in the hexane and ethyl acetate extracts. The flavonoid contents were present at low levels in the aqueous extracts of *C. operculatus* fresh leaves (2.73 mg QE/g) [40] and flower buds (81.1 mg QE/g) [4] when compared with that in the hexane fraction.

In our study, flavonoids have been found to be remarkably rich in the hexane and ethyl acetate fractions. Chalcones belonging to flavonoids family, especially DMC, have been presented as bioactive agents isolated from the non- or low-polar solvent extracts of the flower buds [1,2,9,41,42]. The contents of saponins and tannins were also high in these hexane and ethyl acetate fractions, particularly in the *C. operculatus* flower buds. The tannin content in the hexane fraction was reported to be higher than that in the aqueous extract of *C. nervosum* pulp [43].

It is well known that higher polyphenol and flavonoid contents could significantly increase the antioxidant capacity of plant extracts [44]. In the present study, the hexane and ethyl acetate fractions exhibited weaker antioxidant capacities than the aqueous counterparts. This may be because the content of flavonoids was present at lower levels than those of other phenolic compounds in the *C. operculatus* flower buds and leaves. The aqueous fractions in our study exhibited a similar antioxidant capacity to those of aqueous extracts from *C. operculatus* flower buds and leaves reported by Mai et al. (2009) [37]. Compared to the other studies, most of the *C. operculatus* fractions had a higher antioxidant capacity than those of *Psidium guajava* L. [45], *Halimium halimifolium* [21], and *Syzygium aromaticum* flower buds [22].

The studies by Mai et al. (2009) [37] and Minh et al. (2023) [40] have shown that there were very little to no alkaloids in *C. operculatus*. Our study identified alkaloids in all of the tested fractions at significantly lower concentrations than other phytochemicals. Although the antioxidant activity of alkaloids was also demonstrated [6], the presence of alkaloids at low levels in the fractions of the *C. operculatus* flower buds and leaves could make their contribution secondary when compared with polyphenols and flavonoids.

Among the fractions of *C. operculatus* flower buds and leaves, the hexane fractions and the isolated compound DMC presented the strongest growth inhibitory activity against *H. pylori*. This indicated that flavonoids and other phenolic compounds could have a high synergistic activity against the growth of *H. pylori*. Especially, the single compound DMC, isolated from the flower buds of *C. operculatus*, was found to have two-and-a-half to five times more potent inhibitory activity than these hexane fractions. Similarly, terpenes (e.g., diterpenes and sesquiterpenes) and phenolic compounds from the leaf extract of *Caseria sylvestris* have been reported to provide a high synergism against *H. pylori* [46]. Several studies have shown that extracts from the flower buds and leaves possessed antibacterial activity against many types of Gram-negative and Gram-positive bacteria such as *Xanthomonas* spp. [47], *Staphylococcus aureus*, *S. epidermidis, S. haemolyticus*, *Bacillus subtilis*, *Streptococcus mutans* [6], *H. pylori* [11], and *S. pyogenes* [48]. In addition, DMC has also been known as a bioactive agent against cellular oxidative stress [2] and cytotoxic effects [21,49]. Crude ethanolic extracts from the leaves [11] and crude hexane extracts from the flower buds of *C. operculatus* [4] were previously reported to have high antibacterial activity against *H. pylori*. However, to date, there are no data reported on the antibacterial activity of DMC isolated from *C. operculatus* flower buds against this pathogenic bacterium.

The antibacterial activity of *C. operculatus* fractions against *H. pylori* was found to be higher than those of other reported medicinal plants such as *Cichorium intybus* (MICs of 1.25–10 mg/mL), *Cinnamomum zeylanicum* (MICs of 1.25 to 5 mg/mL), *Foeniculum vulgare* (MICs of >10 mg/mL) [50], and *H. rosa sinensis* (MICs of 0.2–0.25 mg/mL) [28]. It was previously reported that 2′,4′- dihydroxychalcone isolated from leaves of *Muntingia calabura* provided antibacterial activity against methicillin-susceptible and resistant *Staphylococcus aureus* (MICs of 50 and 100 mg/mL, respectively) [51]. Recently, the flavonoids naringenin, myricetin, and luteolin isolated from the red flowers of *H. rosa sinensis* were reported to display effective anti-*H. pylori* properties (MICs of 100–150 μg/mL) [32]. In our study, DMC (2′,4′-Dihydroxy-6′-methoxy-3′,5′-dimethylchalcone) was found to possess potent growth inhibitory activity against *H. pylori* (MICs of 25–50 µg/mL).

*H. pylori* can produce urease for successful survival in acidic conditions and colonization in the gastric mucosa of the human stomach. Therefore, the inhibition of the bacterial enzyme could prevent bacterial growth and colonization. The crude hexane extract from flower buds of *C. operculatus* was found to have a strong inhibitory effect on *H. pylori* urease activity [9] The *C. operculatus* fractions and DMC in our present study have also been found to possess potent inhibitory effects on the urease of *H. pylori*, and is much stronger than those of *Fagonia arabica* L. and *Casuarina equisetifolia* L. reported by Amin et al. (2013) [52]. The flavonoid quercetin, present in the acetone extract of *Heterotheca inuloides* Cass. (Asteraceae), has been known to display a high in vitro inhibition against the enzyme, with an IC_50_ = 132.4 μg/mL [53]. Several other flavonoids (such as naringenin, myricetin, and luteolin) and phenolic acid (protocatechuic acid) from the red flowers of *Hibiscus rosa sinensis* were also reported to produce the potent inhibition of *H. pylori* urease activity [32].

In addition, inhibition of α-amylase and α-glucosidase activities has been known to be one of the treatments for diabetes, since it helps to control glucose levels in the blood. Previously reported by Zhang and Lu (2012) [8], the aqueous extract of *C. operculatus* flower buds and DMC inhibited α-amylase, with IC_50_ values of 73.10 and 20.67 µg/mL, respectively. Recently, Chukiatsiri et al. (2023) [54] indicated that the hexane extract of *C. nervosum* had no inhibitory activity against both α-amylase and α-glucosidase, but the aqueous extract of this plant revealed a depressing effect on both of the enzymes, with IC_50_ values of 0.61 and 0.44 mg/mL, respectively. In the present study, we found that the *C. operculatus* fractions had a strong inhibitory effect on α-glucosidase, while DMC displayed inhibitory activity against α-amylase.

Moreover, adverse effects on the bacterial biofilm formation, cell morphology, and membrane permeability have also been well described as a mode of action of plant secondary metabolites on bacterial survival [55,56]. In the current study, we found that all *C. operculatus* fractions and DMC inhibited urease, with IC_50_ values significantly smaller than their MIC values. At sub-MICs, they also exhibited pronounced antibiofilm activity, as the biofilm growth can act as reservoirs for the spread of the pathogenic bacterium, persistent infection, and resistance to adverse factors [57]. The EtOAc fraction of *H. rosa-sinensis* red flowers at MIC/2 (0.125 mg/mL) has been reported to inhibit the biofilm formation of *H. pylori* by 79.3% and cause the considerable transformation of the spiral forms to the coccoid forms (91 vs. 18% at 1.5 and 0.75 mg/mL after 48 h of treatment) [28]. The flavanone naringenin was found to display the most antibiofilm activity (85.9 versus 52.7% of inhibition at MIC/2 and MIC/4 = 25 μg/mL) [32]. The flavanone also induced the morphological conversion of *H. pylori* to the coccoid forms (95 vs. 16.5% at 1000 and 500 μg/mL), followed by the flavone luteolin (87.5 vs. 14.7%), which caused a greater change to the coccoid forms than that of the flavonol myricetin (79 vs. 15%) [32]. Previously, the methanolic extract of *C. operculatus* leaves were known to inhibit the ability of the acid production and biofilm formation of *Streptococcus mutans*, leading to its anticaries activity [6]. This current study is first report of *anti-H. pylori* and antibiofilm activities and the morphological conversion of the bacterial cells caused by the *C. operculatus* flower bud and leaf fractions and DMC.

Furthermore, naturally occurring chalcones found in many medicinal and edible plants have been known to be precursors of plant flavonoids [58]. Flavonoids possessing antibacterial properties could penetrate the lipid bilayer membrane, causing an increase in membrane permeability and alleviating the bacterial pathogenicity [59]. In our study, the *C. operculatus* flower bud and leaf fractions, especially the isolated DMC, were proven to effectively increase the membrane permeability of *H. pylori*. The results from the SEM images indicated that the hexane fraction of the *C. operculatus* flower buds and DMC produced extensive morphological damage, causing an increase in the membrane permeability, and exerted bactericidal effects. The *H. pylori* cells treated with hesperetin, naringenin, and 7-O-butylnaringenin were shown to be damaged and resulted in morphological alterations or irregular shapes and rough surfaces [60,61]. Hesperidin was reported to interact with bacterial cells and induce membrane disruption, leading to the leakage of cytoplasmic components prior to cell death [62]. Ergüden and Ünver (2021) [63] proposed that phenolic chalcones caused ion leakage from the cytoplasm of Gram-positive bacteria before membrane deformation.

The cytotoxicity experiments showed that the *C. operculatus* fractions and DMC were not toxic to the tested cell lines, in which the flower bud hexane fraction (DMC-rich fraction) was more toxic to the three cancer cell lines (MCF-7, Jurkat, and HeLa) than to the fibroblast cells. The DMC-rich extract obtained from *S. nervosum* fruits was also reported to have stronger anticancer activities against HepG2 (human liver cancer cells) and A549 (human lung cancer cells) than the isolated DMC [21]. These showed that the fractions and DMC have selective inhibitory effects on *H. pylori* urease and induce morphological conversions and membrane disruption, leading to the cell death.

## 5. Conclusions

The fractions from the flower buds and leaves of *C. operculatus* exerted various antioxidant activities. The results demonstrated that the aqueous fractions of the flower buds and leaves, which contain high phenolic contents, exhibited significantly antioxidant effects. The flower bud and leaf hexane fractions, with rich contents of flavonoids and saponins, possessed considerable antibacterial activity towards *H. pylori*. The pronounced anti-*H*. *pylori* activity of the hexane fractions and DMC (2′,4′-dihydroxy-6′-methoxy-3′,5′-dimethylchalcone) may result from its potential role in urease inhibitory and antibiofilm properties. Additionally, the growth-inhibiting and bactericidal effects of the flower bud hexane fractions and DMC have been attributed to causing morphological changes, increasing the permeability, and damaging the cell membrane of *H. pylori*. Moreover, the activities of *C. operculatus* fractions to inhibit α-glucosidase and DMC against α-amylase, and their safety for human cells, indicate the need to evaluate their biological effectiveness in vivo. Flower bud- and leaf-derived materials rich in chalcones and flavonoids, particularly hexane fractions, promise to be a potential source for further pharmaceutical studies.

## Figures and Tables

**Figure 1 biotech-13-00042-f001:**
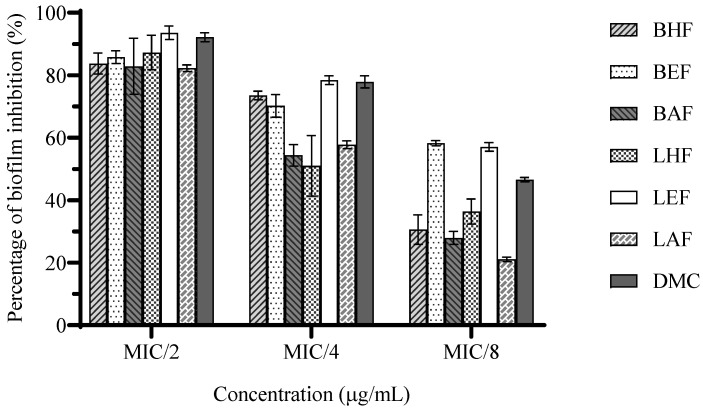
Effect of *C. operculatus* fractions and DMC at sub-MICs on *H. pylori* biofilm formation 48 h post-treatment. Data are reported as means ± SD.

**Figure 2 biotech-13-00042-f002:**
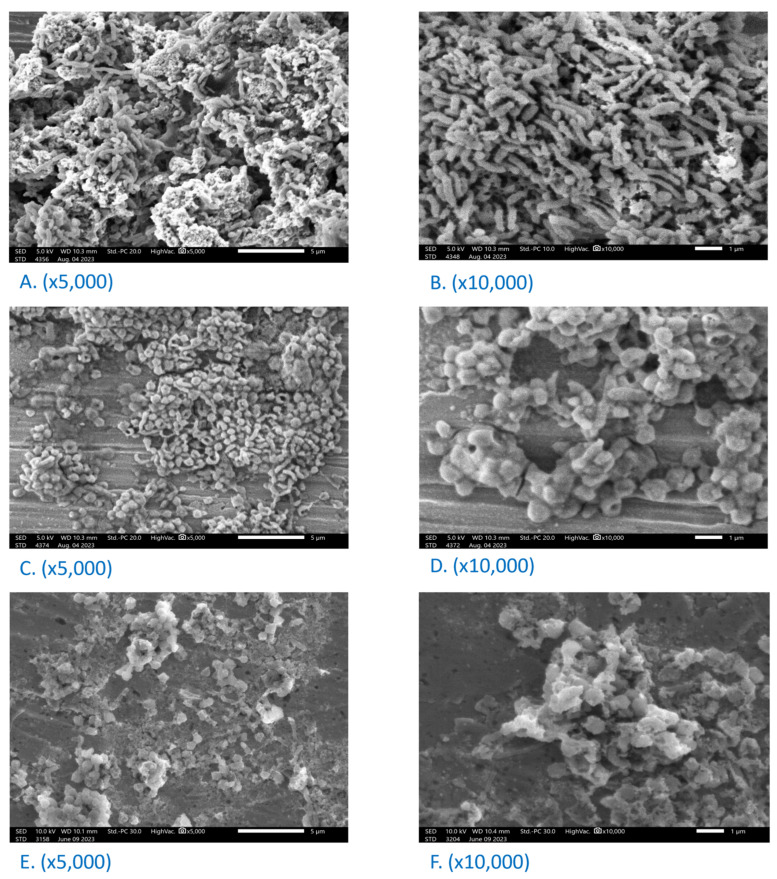
SEM micrographs of *H. pylori* ATCC 43504 depicting untreated cells (**A**,**B**) and cells treated with 125 µg/mL of BHF (**C**,**D**) and 50 µg/mL of DMC (**E**,**F**); bars 5 and 1 µm, respectively.

**Figure 3 biotech-13-00042-f003:**
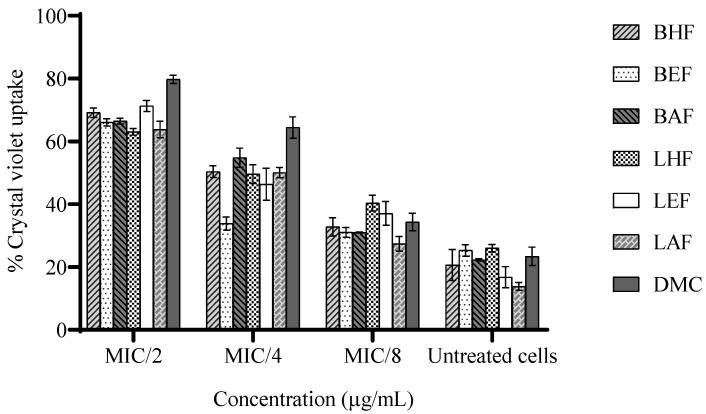
Effect of *C. operculatus* fractions and DMC at sub-MICs on the uptake of crystal violet by *H. pylori* ATCC 43504 after 2 h of treatment. Data are reported as means ± SD (n = 3).

**Table 1 biotech-13-00042-t001:** Total phytochemical contents in different fractions from flower buds and leaves of *C. operculatus*.

Fractions	TPC	TFC	TSC	TTC	TAC
BHF	426.77 ^c^ ± 1.22	134.77 ^a^ ± 7.75	153.33 ^a^ ± 4.69	42.97 ^a^ ± 2.93	1.66 ^d^ ± 0.09
BEF	280.46 ^d^ ± 11.06	85.88 ^b^ ± 2.52	158.10 ^a^ ± 5.97	22.97 ^c^ ± 1.73	1.50 ^d^ ± 0.17
BAF	768.18 ^a^ ± 12.20	11.04 ^e^ ± 0.53	81.59 ^b^ ± 1.20	13.53 ^d^ ± 2.93	3.04 ^c^ ± 0.20
LHF	201.63 ^f^ ± 3.45	76.54 ^bc^ ± 1.72	88.25 ^b^ ± 9.46	33.17 ^b^ ± 2.10	4.81 ^b^ ± 0.09
LEF	238.47 ^e^ ± 4.18	71.72 ^c^ ± 1.74	75.24 ^b^ ± 5.95	10.39 ^d^ ± 1.29	4.73 ^b^ ± 0.05
LAF	490.74 ^b^ ± 7.29	25.56 ^d^ ± 0.21	25.56 ^c^ ± 2.44	4.36 ^e^ ± 0.96	5.41 ^a^ ± 0.08

BHF: flower bud hexane fraction, BEF: flower bud ethyl acetate fraction, and BAF: flower bud aqueous fraction. LHF: leaf hexane fraction, LEF: leaf ethyl acetate fraction, and LAF: leaf aqueous fraction. TPC: total phenolic content (mg GAE/g dry extract), TFC: total flavonoid content (mg QE/g dry extract), TSC: total saponin content (mg OA/g dry extract), TTC: total tannin content (mg CE/g dry extract), and TAC: total alkaloid content (mg AE/g dry extract). Data are shown as mean ± SD (n = 3). Means within a column with different letters indicate a significant difference at *p <* 0.05.

**Table 2 biotech-13-00042-t002:** Antioxidant activity of the different fractions from *C. operculatus* flower buds and leaves.

Samples	FRAP (mg TE/g Extract)	DPPH IC_50_ (µg/mL)	ABTS IC_50_ (µg/mL)
BHF	69.01 ^e^ ± 1.833	33.99 ^c^ ± 0.76	1.70 ^b^ ± 0.09
BEF	91.95 ^c^ ± 1.302	58.46 ^a^ ± 1.370	1.48 ^b^ ± 0.162
BAF	201.80 ^b^ ± 4.502	24.69 ^d^ ± 0.194	1.08 ^c^ ± 0.013
LHF	55.57 ^f^ ± 1.265	39.96 ^b^ ± 0.237	1.06 ^c^ ± 0.084
LEF	78.56 ^d^ ± 1.13	16.05 ^e^ ± 0.031	0.98 ^c^ ± 0.007
LAF	301.82 ^a^ ± 2.306	11.24 ^f^ ±1.524	0.55 ^d^ ± 0.004
Ascorbic acid	ND	3.34 ^g^ ± 0.017	ND
Trolox	ND	ND	2.63 ^a^ ± 0.05

BHF: flower bud hexane fraction, BEF: flower bud ethyl acetate fraction, and BAF: flower bud aqueous fraction. LHF: leaf hexane fraction, LEF: leaf ethyl acetate fraction, and LAF: leaf aqueous fraction. Data are shown as mean ± SD (n = 3). Means within a column with different letters indicate a significant difference at *p <* 0.05. ND: not determined.

**Table 3 biotech-13-00042-t003:** In vitro minimal inhibitory concentration (MIC) values of *C. operculatus* fractions and DMC against *H. pylori.*

Samples	MIC (µg/mL)	
	*H. pylori* ATCC 51932	*H. pylori* ATCC 43504
BHF	125	125
BEF	250	500
BAF	500	500
LHF	125	125
LEF	250	500
LAF	500	1000
DMC	25	50
Amoxicillin	0.01	0.01

BHF: flower bud hexane fraction, BEF: flower bud ethyl acetate fraction, and BAF: flower bud aqueous fraction. LHF: leaf hexane fraction, LEF: leaf ethyl acetate fraction, and LAF: leaf aqueous fraction. Test materials with MIC values of ≤130, >130–<630, 630–1250, >1250–<2500, and ≥2500 µg/mL were classified as the extremely high, high, moderate, low, and no inhibitory activity against the growth of test bacteria, respectively [36].

**Table 4 biotech-13-00042-t004:** Inhibitory effects of *C. operculatus* fractions and DMC on *H. pylori* urease, α-glucosidase, and α-amylase.

Samples	IC_50_ (µg/mL)		
	*H. pylori* Urease	α-Glucosidase	α-Amylase
BHF	2.3 ^e^ ± 0.13	1.5 ^c^ ± 0.09	398.5 ^d^ ± 5.3
BEF	4.9 ^c^ ± 0.40	0.9 ^c^ ± 0.01	497.2 ^b^ ± 17.1
BAF	2.5 ^de^ ± 0.21	0.8 ^c^ ± 0.01	444.3 ^c^ ±14.3
LHF	3.2 ^de^ ± 0.26	2.6 ^c^ ± 0.16	191.3 ^f^ ± 9.5
LEF	3.6 ^d^ ± 0.01	1.2 ^c^ ± 0.06	292.6 ^e^ ± 16.6
LAF	6.8 ^b^ ± 0.07	0.6 ^c^ ± 0.02	1281.7 ^a^ ±23.7
DMC	3.2 ^de^ ± 0.03	94.6 ^a^ ± 2.57	83.80 ^g^ ± 0.08
Thiourea	44.3 ^a^ ± 1.12	ND	ND
Acarbose	ND	25.6 ^b^ ± 0.70	75.0 ^g^ ± 3.86

BHF: flower bud hexane fraction, BEF: flower bud ethyl acetate fraction, and BAF: flower bud aqueous fraction. LHF: leaf hexane fraction, LEF: leaf ethyl acetate fraction, and LAF: leaf aqueous fraction. Data are shown as mean ± SD (n = 3). Means within a column with different letters indicate a significant difference at *p <* 0.05. ND: not determined.

**Table 5 biotech-13-00042-t005:** Cytotoxicity of *C. operculatus* fractions and DMC against four human cell lines.

Samples	CC_50_ (µg/mL)			
	MCF-7	Jurkat	HeLa	Fibroblast
BHF	30.79 ^d^ ± 0.83	18.51 ^d^ ± 1.37	31.70 ^c^ ± 5.74	>100
BEF	57.74 ^c^ ± 1.13	51.06 ^c^ ± 1.42	92.53 ^a^ ± 4.90	>100
BAF	>100	>100	>100	>100
LHF	89.00 ^a^ ± 1.33	56.78 ^c^ ± 2.62	86.45 ^ab^ ± 4.06	>100
LEF	85.43 ^a^ ± 2.76	66.73 ^b^ ± 1.02	91.72 ^a^ ± 1.43	>100
LAF	>100	97.25 ^a^ ± 1.28	77.07 ^b^ ± 5.06	>100
DMC	71.41 ^b^ ± 2.14	73.82 ^b^ ± 7.23	>100	>100
Camptothecin	0.007 ^e^ ± 0.002	0.005 ^e^ ± 0.001	0.89 ^d^ ± 0.088	1.57 ± 0.84

BHF: flower bud hexane fraction, BEF: flower bud ethyl acetate fraction, and BAF: flower bud aqueous fraction. LHF: leaf hexane fraction, LEF: leaf ethyl acetate fraction, and LAF: leaf aqueous fraction. Data are shown as mean ± SD (n = 3). Means within a column with different letters indicate a significant difference at *p <* 0.05.

## Data Availability

The data presented in this study are available in this article and Appendix A.

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
