# Peer review of "Phytochemical Composition, Antioxidant, Anti-Helicobacter pylori, and Enzyme Inhibitory Evaluations of Cleistocalyx operculatus Flower Bud and Leaf Fractions"

_biotech, 2024, doi:10.3390/biotech13040042_

Round 1

Reviewer 1 Report

Comments and Suggestions for Authors

There is no significant interest for this manuscript. The manuscript mainly focused on biological activity evaluation of several plant crude extracts. The phytochemical components were mainly evaluated but as total amount. Only hexane extract was purified to get one compound. The results was just preliminary results and was not qualified enough to published.

Comments on the Quality of English Language

-

Author Response

There is no significant interest for this manuscript. The manuscript mainly focused on biological activity evaluation of several plant crude extracts. The phytochemical components were mainly evaluated but as total amount. Only hexane extract was purified to get one compound. The results was just preliminary results and was not qualified enough to published.

We appreciate the reviewer’s feedback. We think this study makes a valuable contribution to the field of Agricultural and Food Biotechnology. The research shows the potential of flower bud and leaf fractions of Cleistocalyx operculatus and the isolated compound - DMC as antioxidants, and enzyme inhibitory and anti-Helicobacter pylori agents. We hope the paper would be of interest to readers in this area.

Reviewer 2 Report

Comments and Suggestions for Authors

The manuscript shows the potential of bioactive compounds isolated from flower bud and leaf extracts of Cleistocalyx operculatus as antioxidants, anti-Helicobacter pylori, and metabolic syndrome associated enzyme inhibitory activity. The paper is well written and interesting for readers.

Author Response

The manuscript shows the potential of bioactive compounds isolated from flower bud and leaf extracts of Cleistocalyx operculatus as antioxidants, anti-Helicobacter pylori, and metabolic syndrome associated enzyme inhibitory activity. The paper is well written and interesting for readers.

We greatly appreciate your feedback on this manuscript. We believe that these findings will be of useful and interest to the readers of the journal. Thank you very much for your support and attention.

Reviewer 3 Report

Comments and Suggestions for Authors

"Phytochemical composition, antioxidant, anti-Helicobacter pylori, and enzyme inhibitory evaluations of Cleistocalyx operculatus flower bud and leaf fractions"

The manuscript provides a study on the total amounts of groups of bioactive compounds of Cleistocalyx operculatus plant parts, specifically focusing on its antioxidant, antibacterial, and enzyme inhibitory properties. Moreover, the findings have the potential to significantly influence future biomedical research, particularly in the areas of bacterial biofilm inhibition and natural therapies for digestive health. The manuscript could be suitable for publication after revisions. To improve the quality of the manuscript, I have specific questions as outlined below and suggest a few corrections:

·         The accepted plant name is Syzygium nervosum DC., and Cleistocalyx operculatus is a synonym. The accepted plant name should be used in the title of the manuscript.

·         More information about the plant and its traditional medicinal properties could supplement the introduction. Provide additional information to highlight the novelty and relevance of the study.

·         Line 53 unify font style for bacteria.

·         Line 101 lacks reference. Describe the plant material used, the vegetative stage of collection, and the collection site. The study of Thanh et al., 2024 only describes the plant's flower buds, and this study also used the leaves.

·         What concentration of ethanol was used for the initial extraction?

·         Why only FRAP assay was evaluated using Trolox equivalents? Why trolox calibration curve was not applied to ABTS and DPPH assays.?

·         Why did you use ascorbic acid and Trolox for selected assays, not for all?

·         In the conclusions, elucidate which fraction could be the most promising for further research.

Author Response

1, The accepted plant name is Syzygium nervosum DC., and Cleistocalyx operculatus is a synonym. The accepted plant name should be used in the title of the manuscript.

  • Thank you for this suggestion. The plant has been identified as Cleistocalyx operculatus in conventional taxonomic books in Vietnam. The name was also used in many previous researches such as Ye et al. (2024, 2025); Nguyen et al. (2017); Woo et al. (2002); Zhang et al. (2012); Minh et al. (2023), Min et al. (2008); and Bajpai et al. (2010). In addition, the data in the study were used to compared with these researches. Therefore, we have used the plant name Cleistocalyx operculatus in this study.

2, More information about the plant and its traditional medicinal properties could supplement the introduction. Provide additional information to highlight the novelty and relevance of the study.

  • We have added more information about the medicinal properties of Cleistocalyx operculatus in lines 41-44 and 48-51 of the introduction section.

3, Line 53 unify font style for bacteria.

  • Thank you very much. The font style for bacteria at line 58 has been revised.

4, Line 101 lacks reference. Describe the plant material used, the vegetative stage of collection, and the collection site. The study of Thanh et al., 2024 only describes the plant's flower buds, and this study also used the leaves.

  • Thanks for your comment. Information of the operculatus leaves was added and revised in lines 104-111.

5, What concentration of ethanol was used for the initial extraction?

  • In this study, absolute ethanol has been used for crude extract of both flower buds and leaves before doing liquid-liquid fractionation.

6, Why only FRAP assay was evaluated using Trolox equivalents? Why trolox calibration curve was not applied to ABTS and DPPH assays?

  • FRAP assay was performed based on the method of Rebaya et al. (2015), in which the results of antioxidant activity were expressed in mg Trolox equivalent (TE)/g, so Trolox was used to measure the standard curve. The DPPH and ABTS methods were performed according to the studies of Elouafy et al. (2023) and Olszowy-Tomczyk and Typek (2024), respectively.

7, Why did you use ascorbic acid and Trolox for selected assays, not for all?

  • We have followed on different references to evaluate the antioxidant activity of the samples, leading to the difference in reference compounds. Trolox was used as the positive control for the ABTS assay by Olszowy-Tomczyk and Typek (2024) because it reacts quickly and has high stability after the reaction time (Kuts et al., 2022). And ascorbic acid was used as the reference control in DPPH assay by Rebaya et al. (2015).

8, In the conclusions, elucidate which fraction could be the most promising for further research.

Thank you very much for your advices. The phrase “particularly hexane fractions” was added in the last sentence (line 616) of the conclusion section.

Round 2

Reviewer 3 Report

Comments and Suggestions for Authors

The authors have addressed the reviewer's concerns. Grammatical mistake in line 619 "researchs " should be "researches".